



# Large seasonal and interannual variations of biogenic sulfur compounds in the Arctic atmosphere (Svalbard; 78.9° N, 11.9° E)

Sehyun Jang[1,*], Ki-Tae Park[2,3,*], Kitack Lee[1], Young Jun Yoon[2], Kitae Kim[2,3],
Hyun Young Chung[2,3], Silvia Becagli[4], Bang Yong Lee[2], Rita Traversi[4],
Konstantinos Eleftheriadis[5], Radovan Krejci[6], Ove Hermansen[7]

[1]Division of Environmental Science and Engineering, Pohang University of Science and Technology, Pohang, 37673, Korea
[2]Korea Polar Research Institute (KOPRI), 26 Songdomirae-ro, Yeonsu-gu, Incheon, 21990, Korea
[3]University of Science and Technology, Daejeon, 34113, Korea
[4]Institute of Polar Science, ISP-CNR, via Torino, 155, Venezia Mestre, (VE) 30172, Italy
[5]N.C.S.R. "Demokritos", Environmental Radioactivity Laboratory, Ag. Paraskevi, 15341, Attiki, Greece
[6]Department of Environmental Science and Analytical Chemistry & Bolin Centre for Climate Research, Stockholm University, Stockholm 10691, Sweden
[7]Norwegian Institute for Air Research, Kjeller, Norway

[*]These authors contributed equally to this work.

*Correspondence to*: Kitack Lee (ktl@postech.ac.kr) and Young Jun Yoon (yjyoon@kopri.re.kr)

**Abstract.** Seasonal to interannual variations in the concentrations of sulfur aerosols (< 2.5 μm in diameter; non sea-salt sulfate: $NSS\text{-}SO_4^{2-}$; anthropogenic sulfate: $Anth\text{-}SO_4^{2-}$; biogenic sulfate: $Bio\text{-}SO_4^{2-}$; methanesulfonic acid: MSA) in the Arctic atmosphere were investigated using measurements of the chemical composition of aerosols collected at Ny-Ålesund, Svalbard (78.9° N, 11.9° E) from 2015 to 2019. In all measurement years the concentration of $NSS\text{-}SO_4^{2-}$ was highest during the pre-bloom period and rapidly decreased towards summer. During the pre-bloom period we found a strong correlation between $NSS\text{-}SO_4^{2-}$ and $Anth\text{-}SO_4^{2-}$ because more than 50 % of the $NSS\text{-}SO_4^{2-}$ measured during the pre-bloom period was $Anth\text{-}SO_4^{2-}$, which originated in the northern Europe and was subsequently transported to the Arctic through the Arctic haze. Unexpected increases in the concentration of $Bio\text{-}SO_4^{2-}$ aerosols (an oxidation product of dimethylsulfide: DMS) were occasionally found during the pre-bloom period and were obviously not produced in ocean areas in the proximity of Ny-Ålesund, but probably originated in distant regions to the south (i.e., the North Atlantic Ocean and the Norwegian Sea). The concentration of MSA (another oxidation product of DMS) during the pre-bloom period contrarily remained low, which was largely because of the greater loss of MSA relative to $Bio\text{-}SO_4^{2-}$ and the suppression of condensation of gaseous MSA onto existing particles during the northward transport of air masses containing these components from distant ocean source regions. Moreover, the low light intensity during the pre-bloom period resulted in a low concentration of photochemically activated oxidant species including OH radicals and BrO and thus more favoured the oxidation pathway of DMS to $Bio\text{-}SO_4^{2-}$ rather than to MSA, which acted to lower the MSA concentration at Ny-Ålesund. The concentration of MSA peaked in May or June, and was positively correlated with ocean biomass in the Greenland and Barents seas around Svalbard. As a





result, the mean ratio of MSA to the DMS-derived aerosols was low ($0.09 \pm 0.07$) for the pre-bloom period but high ($0.32 \pm 0.15$) for the bloom and post-bloom periods. Our results indicate that the contribution of MSA to the growth of the newly formed particles to a size at which they could act as condensation nuclei was considerably greater during the bloom and post-bloom periods than during the pre-bloom period.

## 1. Introduction

Aerosols alter the radiative properties of the Earth's surface by means of direct (e.g., scattering and absorption of solar radiation) and indirect (e.g., cloud life-time) effects, and thereby contribute to climate change (Albrecht, 1989; Haywood and Boucher, 2000; Sekiguchi et al., 2003). Moreover, acidification of the Arctic Ocean has been enhanced because of increasing addition of anthropogenic $CO_2$ facilitated by ocean freshening and greater air-sea $CO_2$ exchange (Lee et al., 2011). The recent acceleration of Arctic warming has highlighted the role of natural aerosols in influencing the

radiative properties of the Arctic atmosphere (Dall'Osto et al., 2017; Willis et al., 2018). Nonetheless, current knowledge of the effect of aerosols on climate regulation and the mechanisms of formation of natural aerosols is far from comprehensive, and more alarmingly is ambiguous (Mahowald et al., 2011; IPCC, 2013). In the atmosphere sulfate aerosols effectively form new particles through homogeneous nucleation and clustering reactions that are closely linked to water vapor and ammonia (negative ion-induced ternary nucleation; Kulmala, 2003; Kulmala et al., 2004), whereas methanesulfonic acid (MSA)

particles tend to condense more onto existing particles and eventually contribute to particle growth (Wyslouzil, et al., 1991; Leaitch et al., 2013; Hayashida et al., 2017). The production of sulfuric acid and MSA (having low volatility vapors) when few condensation sites are available could result in an increase in new particle formation (NPF) (Boy et al., 2005), so dimethylsulfide (DMS) emissions from the Arctic Ocean could contribute considerably to NPF and climate regulation (Dall'Osto et al., 2017). The growth of particles following NPF is particularly crucial in generating cloud condensation

nuclei (CCN), which eventually lead to cloud formation. As a result, naturally produced gas molecules can promote NPF and subsequent growth of particles in the presence of sulfate and MSA (DMS oxidation products) (Chang et al., 2011a; Burkart et al., 2017). Hence, data on the quantities of non sea-salt sulfate ($NSS-SO_4^{2-}$) and MSA and their variations are crucial in elucidating NPF and particle growth, and ultimately the role of ocean phytoplankton in modulation of the radiative properties of the Arctic atmosphere.

The origins of sulfate aerosols include sea-salt sulfate ($ss-SO_4^{2-}$), anthropogenic $SO_2$, volcanic $SO_2$, boreal production of natural precursor, and DMS (Bates et al., 1992a). Among those, DMS is only produced in the upper ocean by means of multiple biological processes (e.g., Kettle and Andreae, 2000; Stefels et al., 2007; Kim et al., 2010; Lee et al., 2012; Park et al., 2014a). Some of the DMS is ultimately released into the atmosphere through air-sea gas exchange processes. Airborne DMS is rapidly oxidized to $SO_2$ via hydrogen abstraction by OH radicals, nitrate, and chlorine; to

hydroperoxymethyl thioformate via hydrogen shift by OH radicals; and to MSA via OH addition by OH radicals and in part by halogen oxides (von Glasow and Crutzen, 2004; Barnes et al., 2006; Veres et al., 2020). Seasonal variations in the



product ratio of DMS oxidized to MSA and biogenic sulfate (Bio-SO$_4^{2-}$) over the Arctic region reflect the complexity of aerosol chemistry. The product ratio of DMS oxidation is highly variable, and is affected by air temperature, relative humidity, precipitation, and solar radiation (Hynes et al., 1986; Yin et al., 1990; Bates et al., 1992b). Among those factors

involved, air temperature is known to largely determine the oxidation pathways of DMS. At ambient temperatures the proportions of MSA and Bio-SO$_4^{2-}$ are typically 0.25 and 0.75, respectively (Hynes et al., 1986). DMS is well known to be oxidized more to MSA at lower temperatures. The observed latitudinal variations in the product ratio of DMS oxidation are largely consistent with those predicted from the temperature dependence of the oxidation pathway of DMS (Hynes et al., 1986; Berresheim et al., 1990; Bates et al., 1992b), although equally available are reports on an absence of temperature

dependence (Ayers et al., 1991; Prospero et al., 1991; Chen et al., 2012). The product ratio of DMS oxidation is a result of the net effect of multiple processes, including concentration of atmospheric oxidants and meteorological factors influencing DMS oxidation. Therefore, the ratio could vary considerably among seasons and years.

To investigate DMS oxidation pathways in the Arctic atmosphere we measured sulfate aerosol concentrations at 3-day intervals from 2015 to 2019; this provided comprehensive datasets encompassing seasonal and interannual variations in

sulfate and MSA concentrations in aerosol particles in the Arctic atmosphere. In particular, S isotope ratios were measured for all aerosol samples, and were used to partition the total NSS-SO$_4^{2-}$ into anthropogenic sulfate (Anth-SO$_4^{2-}$) and Bio-SO$_4^{2-}$ (the oxidative product of biogenic DMS). We also calculated the product ratio of MSA to biogenic aerosols (MSA + Bio-SO$_4^{2-}$: Bio-aerosol). Analysis of Anth-SO$_4^{2-}$, Bio-SO$_4^{2-}$, and MSA concentration data, in conjunction with data on air mass back-trajectories enabled identification of the sources of S aerosols, and elucidation of factors governing variations in

their concentrations.

## 2. Materials and methods

### 2.1. Sampling site and aerosol sampling

Aerosol samples were collected at 50 m above sea level at the Gruvebadet observatory (78.9° N, 11.9° E; Fig. 1a) at Ny-Ålesund, Svalbard. Sampling covered the phytoplankton pre-bloom (defined as March to the 2nd week of April), bloom

(3rd week of April to the 2nd week of June), and post-bloom periods (3rd week of June onwards). Division of these periods was subjectively made based on the mean chlorophyll-*a* (Chl-*a*) concentration in the Greenland and Barents seas near Svalbard. The period during which the concentration of Chl-*a* was > 0.5 mg m$^{-3}$ was defined as the phytoplankton bloom period, whereas the periods when the concentration of Chl-*a* was < 0.5 mg m$^{-3}$ prior to and following the bloom were defined as the pre-bloom and post-bloom periods, respectively.

Aerosol samples were collected at 3-day intervals using a high volume sampler (HV-1000R; SIBATA, Japan) outfitted with a PM 2.5 impactor (collecting particles < 2.5 µm in aerodynamic equivalent diameter). The aerosol sampler was mounted on the roof of the Gruvebadet observatory. Particulate matter in the atmosphere was collected on a quartz filter



over approximately 72 h at a flow rate of 1000 L min$^{-1}$, corresponding to a total air volume of 4320 m$^3$. The method of aerosol sampling has been described elsewhere (Park et al., 2017).

**2.2. Atmospheric DMS mixing ratio and major ions in aerosol samples**

The analytical system enabling measurement of atmospheric DMS mixing ratio at parts per trillion levels is equipped with a DMS trapping component, a gas chromatograph, and a pulsed flame photometric detector. The detection limit of the DMS system was close to 1.5 pptv with a sampling air volume of 6 L and the description of the system can be found elsewhere (Jang et al., 2016).

For determination of concentrations of major ions, a disk filter (47-mm diameter) was taken from a whole quartz filter (20.3 cm × 25.4 cm), soaked in 50 mL of Milli-Q water and sonicated in a bath for 60 min; aliquots of this solution were used for analysis. Milli-Q water used for the ion extraction was produced using a water purification system (Milli-Q Direct 16, Merck Millipore, USA). The concentrations of water-extractable inorganic anions and cations including MSA, were measured using ion chromatography (Dionex ICS-1100, Thermo Fisher Scientific Inc., USA) fitted with an IonPac AS

19 column (Thermo Fisher Scientific Inc., USA). The instrumental detection limits were 0.02 µg L$^{-1}$ for MSA and 0.02 µg L$^{-1}$ for SO$_4^{2-}$. From replicate injections, the analytical precision was determined to be < 5 % (relative standard deviation).

**2.3. Stable S isotope ratio in sulfate aerosols**

For measurement of stable S isotope ratio ($\delta^{34}$S) in an aerosol sample, half of the quartz filter was soaked in 50 mL Milli-Q water and sonicated for 60 min. Then, 50–100 µL of 1 M HCl was added to the solution (resulting in a pH of 3–4),

after which 100 µL of 1 M BaCl$_2$ solution was injected into the solution, leading to gradual precipitation of BaSO$_4$. Following the completion of precipitation over 24 h, the BaSO$_4$ precipitates were filtered onto a membrane filter and dried for another 24 h prior to S isotope ratio measurement. Each membrane filter was packed into a tin capsule and analyzed using an isotope ratio mass spectrometer (IsoPrime100; IsoPrime Ltd, UK) and an elemental analyzer (Vario MICRO cube; Elementar Co., Germany). Each filter treatment was carried out in a laminar flow hood to minimize contamination.

International standard reference materials were used to measure the abundance of S isotope in the aerosols. We used NBS-127 (20.3 ± 0.4 ‰), IAEA-S1 (silver sulfide; -0.3 ± 0.3 ‰), and IAEA-S2 (silver sulfide; 22.7 ± 0.2 ‰) (Coplen and Krouse, 1998; Halas and Szaran, 2001; Santamaria-Fernandez et al., 2008) to prepare the calibration curve. NBS-127 was used as the primary standard reference material, and was measured with every five samples.

The resulting S isotope ratio of an aerosol sample ($\delta^{34}$S) was expressed (Eq. 1) as parts per thousand (‰) relative to

the $^{34}$S/$^{32}$S ratio of a standard (Vienna-Canyon Diablo Troilite) (Krouse and Grinenko, 1991).

$$\delta^{34}S \ (‰) = \{(^{34}S/^{32}S)_{sample}/(^{34}S/^{32}S)_{standard} - 1\} \times 1000 \qquad (1)$$





Among known sources, both Anth-$SO_4^{2-}$ and Bio-$SO_4^{2-}$ are the main sources of sulfate aerosols in the Arctic environment

(Udisti et al., 2016; Park et al., 2017). Data on the S isotope ratio of aerosol particles and the concentrations of major ions

enabled estimation of the contributions of biogenic DMS ($f_{bio}$), anthropogenic $SO_x$ ($f_{anth}$), and ss-$SO_4^{2-}$ ($f_{ss}$) to the total $SO_4^{2-}$

concentration. The concentration of ss-$SO_4^{2-}$ was estimated using the seawater ratio of $SO_4^{2-}$ to $Na^+$ (0.252; Keene et al.,

1986). The NSS-$SO_4^{2-}$ fraction of the total $SO_4^{2-}$ was then calculated by subtracting the fraction of ss-$SO_4^{2-}$ from the total

$SO_4^{2-}$. The fraction of biogenic $SO_4^{2-}$ was estimated by solving the following equations:


$$f_{anth} + f_{bio} + f_{ss} = 1 \qquad (2)$$

$$\delta^{34}S_{sample} = f_{anth}\delta^{34}S_{anth} + f_{bio}\delta^{34}S_{bio} + f_{ss}\delta^{34}S_{ss} \qquad (3)$$

$$f_{ss} = [SO_4^{2-}/Na^+]_{ss} \cdot [Na^+/SO_4^{2-}]_{sample} \qquad (4)$$

To solve equations 2–4 we used the reported S isotope ratios ($\delta^{34}S$) of ss-$SO_4^{2-}$ (21.0 $\pm$ 0.1 ‰), Anth-$SO_4^{2-}$ (5 $\pm$ 1 ‰), and

Bio-$SO_4^{2-}$ (18 $\pm$ 2 ‰) (Norman et al., 1999; Böttcher et al., 2007; Lin et al., 2012). Based on measurements of the S isotope

ratio on aerosol samples, then we calculated the fraction of MSA ($R_{Bio}$ = MSA / [MSA + Bio-$SO_4^{2-}$]) in the total biogenic

aerosols to evaluate the oxidative pathway of DMS to MSA or to Bio-$SO_4^{2-}$. In calculating $R_{Bio}$, some data (~23 data) having

low Bio-$SO_4^{2-}$ values (< 25 ng m$^{-3}$) were not included because unusually low Bio-$SO_4^{2-}$ values resulted in biases in the $R_{Bio}$

values (Table S1).

### 2.4. Black carbon

An aethalometer (model AE31; Magee Scientific Co., USA) installed at the Zeppelin station was used to analyze

the concentration of equivalent black carbon by measuring light-absorbing particles at a wavelength of 880 nm, as described

by Eleftheriadis et al. (2009). The good congruence between the concentrations of Anth-$SO_4^{2-}$ and black carbon measured

during the pre-bloom period (March to April) indicates that variations in black carbon were reasonably consistent with

variations in Anth-$SO_4^{2-}$, reflecting that both Anth-$SO_4^{2-}$ and black carbon had common sources (i.e., fossil fuel combustion

and forest burning) (Text S1; Figs. S1 and S2).

### 2.5. Air mass origin, chlorophyll-*a* concentration, and meteorological parameters

Both 8-day and monthly mean Chl-*a* concentration data level-3 MODIS Aqua were downloaded from the NASA

OCEAN Color website (http://oceancolor.gsfc.nasa.gov/) at a 4-km resolution. The three-dimensional 5-d (120 h) back

trajectories were calculated using the Hybrid Single-Particle Lagrangian Integrated Trajectory model from the NOAA Air

Resources Laboratory (Draxler and Hess, 1998). Meteorological parameters including solar radiation, relative humidity, and

air temperature at each time point were also calculated along the air mass trajectories. The calculations were made based on





meteorological data from Global Data Assimilation System (at 1° latitude × 1° longitude resolution) produced by the

National Centers for Environmental Prediction. Air masses were modelled to arrive at an altitude of 50 m above sea level at

the Gruvebadet station at each hour of the study period. To identify the major air mass pathways prior to reaching to the

Gruvebadet station, the calculated air mass trajectories were grouped into several clusters using the k-means clustering

algorithm. Monthly mean air temperature data at 900 hPa were obtained from European Centre for Medium-Range Weather

Forecasts Reanalysis 5 at a 30-km resolution. Sea level pressure data were obtained from the National Oceanic and

Atmospheric Administration Physical Sciences Laboratory (http://psl.noaa.gov/).

## 3. Results

### 3.1. Atmospheric DMS mixing ratio

The mixing ratio of atmospheric DMS, a precursor of Bio-$SO_4^{2-}$ and MSA, showed considerable (several orders of

magnitude) variability at daily to weekly intervals during the bloom and post-bloom periods (Figs. 2a and 3b). The

atmospheric DMS mixing ratio generally corresponded to the phytoplankton biomass in the oceans surrounding Svalbard

(Figs. 2a and S3). During the bloom period the maximum monthly mean mixing ratio of DMS occurred in May 2015 (68.4 $\pm$

86.8 pptv); an increase in the DMS mixing ratio continued until August of that year, reflecting the persistent phytoplankton

biomass producing DMS in the vicinity of Svalbard. Based on our atmospheric DMS concentration data, we conclude that

DMS was ubiquitous in the Arctic atmosphere for the entire period of warming from the phytoplankton bloom to post-bloom

periods (Park et al., 2013).

### 3.2. S isotopic composition ($\delta^{34}$S) and sources of sulfate aerosols

The $\delta^{34}$S values for sulfate aerosols ranged from 2.2 to 17.6 ‰ between March and August (Fig. 2b). In all years of

measurement, the $\delta^{34}$S values were low in April or earlier months, rapidly increased towards May to June, and remained high

towards August (Fig. 3c). As warming progressed, the trend of increasing $\delta^{34}$S in the sulfate aerosols was broadly consistent

with the increasing mixing ratio of atmospheric DMS. The $\delta^{34}$S values for the pre-bloom, bloom, and post-bloom periods

averaged over five years were 7.5 $\pm$ 2.6 ‰, 9.5 $\pm$ 2.8 ‰, and 11.3 $\pm$ 2.8 ‰, respectively, reflecting an increasing

enrichment in the heavier $^{34}$S towards summer. The maximum monthly mean $\delta^{34}$S (13.5 $\pm$ 2.6 ‰) occurred in July 2018,

whereas the lowest mean (3.7 $\pm$ 1.8 ‰) occurred in April 2019. The mean pre-bloom $\delta^{34}$S value in 2017 (9.2 $\pm$ 1.8 ‰) was

higher than in 2018 (5.9 $\pm$ 1.2 ‰), whereas the mean bloom and post-bloom $\delta^{34}$S values were marginally lower in 2017

(11.0 $\pm$ 2.0 ‰) than in 2018 (12.5 $\pm$ 2.8 ‰).

On monthly scales the greatest contribution of Bio-$SO_4^{2-}$ occurred in August 2018 (59.4 $\pm$ 17.2 %) (Fig. S4). The

proportion of Bio-$SO_4^{2-}$ among all $SO_4^{2-}$ particles was 18.1 $\pm$ 16.6 % during the pre-bloom period, and then sharply





increased to 37.2 $\pm$ 21.0 % during the bloom and post-bloom periods, whereas the contribution of anthropogenic $SO_2$ was 79.2 $\pm$ 16.9 % during the pre-bloom period and 57.9 $\pm$ 21.4 % during the bloom and post-bloom periods. It is surprising that

Anth-$SO_4^{2-}$ was found to be the largest contributor to total $SO_4^{2-}$ during all three periods (Fig. S4).

### 3.3. NSS-$SO_4^{2-}$, Anth-$SO_4^{2-}$, and Biogenic sulfur aerosols

There were considerable seasonal and interannual variations in the concentrations of S aerosols including NSS-$SO_4^{2-}$, Anth-$SO_4^{2-}$, and Bio-aerosol (Figs. 3d–h, 4 and 5). In all years of the study the seasonal mean NSS-$SO_4^{2-}$ concentration reached a maximum during the pre-bloom period (857 $\pm$ 520 ng m$^{-3}$), decreased rapidly towards summer, and

eventually dropped to a quarter of the maximum value during the post-bloom period (212 $\pm$ 120 ng m$^{-3}$). We also found that the NSS-$SO_4^{2-}$ concentration in the months prior to May varied by as much as a factor of three (1015 $\pm$ 586 in 2015 versus 291 $\pm$ 93 ng m$^{-3}$ in 2019). The highest monthly mean NSS-$SO_4^{2-}$ concentration (1309 $\pm$ 131 ng m$^{-3}$) was recorded in March 2017, and the lowest was in July 2018 (165 $\pm$ 128 ng m$^{-3}$). The concentration of Anth-$SO_4^{2-}$ showed a temporal trend similar to that of NSS-$SO_4^{2-}$, with the highest monthly mean concentration (678 $\pm$ 450 ng m$^{-3}$) occurring during the pre-

bloom period, followed by a trend of decrease for the bloom (369 $\pm$ 236 ng m$^{-3}$) and post-bloom (114 $\pm$ 78 ng m$^{-3}$) periods.

During the pre-bloom period, when the phytoplankton biomass was nearly absent in waters around Svalbard, the concentration of Bio-$SO_4^{2-}$ was unexpectedly high (180 $\pm$ 213 ng m$^{-3}$), reaching 743 ng m$^{-3}$ in 2016 (Fig. 4c). During the phytoplankton bloom period, the seasonal mean concentration of Bio-$SO_4^{2-}$ was highest (184 $\pm$ 190 ng m$^{-3}$). As summer approached, the Bio-$SO_4^{2-}$ concentration decreased slightly during the post-bloom periods (98 $\pm$ 68 ng m$^{-3}$; Fig. 4c). In

contrast to the trend for Bio-$SO_4^{2-}$, the MSA concentration remained low (< 30 ng m$^{-3}$) during the pre-bloom period, and rapidly increased during the transition from the pre-bloom to bloom periods (Figs. 3g and 5a). An elevated MSA concentration maintained during much of the bloom and post-bloom periods, and then it decreased slightly to near the detection limit by the end of August. The highest monthly mean MSA concentrations were found in May (81.4 $\pm$ 58.1 ng m$^{-3}$) and June (81.9 $\pm$ 56.5 ng m$^{-3}$), which broadly agree with previous MSA measurements at Svalbard (Becagli et al.,

2019). The annual mean concentrations of MSA (March to August) varied slightly among years (46.2 $\pm$ 35.9 ng m$^{-3}$ in 2017, 63.5 $\pm$ 52.9 ng m$^{-3}$ in 2018, and 55.4 $\pm$ 45.5 ng m$^{-3}$ in 2019). The Bio-aerosol concentration increased with the onset of the spring bloom, and stayed at moderate levels until June (Figs. 3h and 5b). The concentration of Bio-aerosol during the bloom period (252 $\pm$ 197 ng m$^{-3}$) was slightly higher than that during post-bloom period (149 $\pm$ 91 ng m$^{-3}$), and the highest monthly concentration of Bio-aerosol was found in April or May in all measurement years. The total concentrations of Bio-

aerosol during the bloom and post-bloom periods were comparable in all three years (214 $\pm$ 124 ng m$^{-3}$ in 2017, 204 $\pm$ 174 ng m$^{-3}$ in 2018, and 160 $\pm$ 153 ng m$^{-3}$ in 2019).





### 3.4. Ratio of MSA to Bio-aerosol ($R_{Bio}$)

In all years of this study the $R_{Bio}$ values derived from $\delta^{34}$S data were lowest during the pre-bloom period and

increased in the transition to the spring bloom, as biogenic DMS production peaked (Figs. 3i and 5c). The $R_{Bio}$ value varied

by a factor of three over seasons, showing maximum values during the bloom period ($0.32 \pm 0.17$), and lowest values during

the pre-bloom period ($0.09 \pm 0.07$). The highest mean $R_{Bio}$ ($0.49 \pm 0.05$) was found in June 2018, whereas the lowest $R_{Bio}$

($0.08 \pm 0.01$) was found in March 2017. There were large interannual variations in the seasonal mean $R_{Bio}$ ($0.24 \pm 0.11$ in

2017, $0.40 \pm 0.14$ in 2018, and $0.36 \pm 0.14$ in 2019) during the bloom and post-bloom periods.

Similar $R_{Bio}$ values were also reported at Ny-Ålesund. For example, Udisti et al (2016) reported a MSA to Bio-

$SO_4^{2-}$ ratio of 0.33 ($R_{Bio} = 0.25$) during the spring-summer period in 2014. This ratio was derived from a multi-seasonal

asymptotic value in a plot between the MSA to NSS-$SO_4^{2-}$ ratio and the MSA concentration. Implicit in this calculation is

the assumption that the fraction of Bio-$SO_4^{2-}$ in the total NSS-$SO_4^{2-}$ aerosols is overwhelming when the MSA to NSS-$SO_4^{2-}$

ratio approaches the asymptotic value (Udisti et al., 2016; Park et al., 2017). Other investigators also reported comparable

$R_{Bio}$ values in other Arctic environments: 0.18–0.20 at the central Arctic Ocean (Chang et al., 2011b; Leck and Persson

1996); 0.28 at the Eastern Antarctic Plateau (Udisti et al., 2012); and 0.28 at Alert (Norman et al., 1999). These $R_{Bio}$ values

were all derived from a multi-seasonal asymptotic value in a plot between the MSA to NSS-$SO_4^{2-}$ ratio and MSA

concentration. The analytical accessibilities associated with measurements of MSA and NSS-$SO_4^{2-}$ concentration (i.e., less

laborious and requires fewer aerosols than is needed for the technique measuring the S-isotope ratio) make data on the MSA

to NSS-$SO_4^{2-}$ ratio more widely available.

## 4. Discussion

### 4.1. Factors affecting variations in the S aerosol concentration in the Arctic atmosphere

Seasonal variations in NSS-$SO_4^{2-}$ aerosols were strongly associated with variations in Anth-$SO_4^{2-}$. In particular, the

tight association of these parameters indicates that Anth-$SO_4^{2-}$ aerosols were the largest contributor to NSS-$SO_4^{2-}$ during the

pre-bloom period, when the intrusion of Arctic haze is considerable (Figs. S4 and S5). During the transition from the pre-

bloom to bloom periods, the input of Anth-$SO_4^{2-}$ particles to our study area rapidly decreased because of weakening of the

northward transport of air masses (containing Anth-$SO_4^{2-}$) from Europe and increasing removal of Anth-$SO_4^{2-}$ aerosols by

increasing precipitation as the seasons progress (Li and Barrie, 1993) (Fig. 4b). The decreasing input of Anth-$SO_4^{2-}$ particles

to the observation site during the bloom and post-bloom periods was also independently confirmed by the trend of decrease

in the measured black carbon concentration at our observation site (Figs. 3a and S2).



The large interannual variability in NSS-SO$_4^{2-}$ from March to April was strongly associated with changes in the trajectory of the air masses reaching Svalbard, and the sea level pressure along those air mass trajectories (Fig. 6). More explicitly, the higher concentrations of NSS-SO$_4^{2-}$ particles in 2015 (1015 $\pm$ 586 ng m$^{-3}$) resulted from the greater input of pollutants (Anth-SO$_4^{2-}$) from northern Europe via the intensified south-westerly wind, whereas the opposite occurred in 2018 and 2019 (634 $\pm$ 266 for 2018 and 291 $\pm$ 93 ng m$^{-3}$ for 2019).

An unusual elevation of the Bio-SO$_4^{2-}$ concentration was occasionally found during the pre-bloom periods in 2016 and 2017, despite the absence of biological activity in the sea ice-covered oceans surrounding Svalbard (Figs. 4c and S6). The spikes in the Bio-SO$_4^{2-}$ concentration likely originate from Bio-SO$_4^{2-}$ aerosols that were produced in distant ocean regions (e.g., the North Atlantic Ocean, the Norwegian Sea, and further south of 50° N–70° N and 25° W–50° E), and then carried into the Arctic via a northward transport of air masses. Analysis of air mass back trajectory data showed that the elevated values of Bio-SO$_4^{2-}$ during the pre-bloom period in 2016 and 2017 resulted from air masses from lower latitude regions reaching Svalbard, rather than originating locally from the oceans around Svalbard, while the much lower Bio-SO$_4^{2-}$ concentrations in 2018 probably resulted from an absence of air masses originating from distant DMS source regions during the pre-bloom period (Figs. 4c and S7).

The MSA concentration remained low during the pre-bloom period (i.e., no apparent high peaks), largely because of the greater removal of MSA relative to Bio-SO$_4^{2-}$ aerosols during long-range transport to Svalbard from the distant source regions to the south. For example, MSA tends to more easily condense onto existing particles (Hoppel, 1987; Pszenny et al., 1989) because of its higher vapor pressure, and is thus more rapidly removed from the atmosphere with larger particles through wet deposition; this results in greater loss of MSA relative to SO$_4^{2-}$. Greater enrichment of MSA occurs in super-micron sized particles than in submicron particles (Legrand and Pasteur, 1998). The higher ratios of MSA to NSS-SO$_4^{2-}$ in rainwater and fresh snows than in aerosol particles is also indicative of the greater removal of MSA (Berresheim et al., 1991; Jaffrezo et al., 1994). The production mechanism of MSA (via DMS oxidation by OH radicals) (Gondwe et al., 2004) could also lower the MSA concentration during the pre-bloom period, when the low levels of OH radicals (as a result of low light conditions) resulted in less MSA production. The elevations of MSA occurred in May or June, when the production of OH radicals was high and associated with increasing solar radiation and biological production (Fig. S8 and S9a).

The concentrations of Bio-aerosol during the bloom and post-bloom periods were comparable in all three years (214 $\pm$ 124 ng m$^{-3}$ in 2017, 204 $\pm$ 174 ng m$^{-3}$ in 2018, and 160 $\pm$ 153 ng m$^{-3}$ in 2019), despite differing phytoplankton biomass (derived from Chl-$a$) among those years (Fig. S9b). This mismatch has been reported previously, and suggests that estimations of marine organic aerosols based on Chl-$a$ data only are unreliable (Rinaldi et al., 2013). In particular, the summer DMS-driven aerosols produced from the Barents Sea were not proportional to the Chl-$a$ concentrations (Becagli et al., 2016). Different compositions of phytoplankton species in different ocean domains (Greenland Sea versus Barents Sea) could also result in changes in DMS production because phytoplankton have differing cellular levels of dimethylsulfoniopropinate (DMSP; a precursor of DMS) and the DMSP cleavage enzyme (enabling the transformation of



DMSP to DMS) (Park et al., 2014b). The DMS production capacity in the Greenland Sea (where prymnesiophytes dominate)
was found to be 3-fold higher than that in the Barents Sea (where diatoms dominate) (Park et al., 2018). Other studies have
also reported that the concentrations of MSA or Bio-$SO_4^{2-}$ do not always follow the atmospheric DMS mixing ratio,
highlighting the involvement of other factors in the oxidation of DMS to MSA or Bio-$SO_4^{2-}$ (Read et al., 2008; Yan et al.,
2020a). Therefore, the amounts of DMS produced and its oxidation products may not be solely explained by variations in the
ocean biomass.

In the Arctic summer atmosphere the low abundance of large particles (i.e., Aitken and accumulation mode) could
probably enhance the formation of new particles via the gas-to-particle conversion process and the ultimate initiation of
CCN formation (Boy et al., 2005; Dall'Osto et al., 2018). The concurrent increase in biogenic sulfate aerosols and small
sized particles (3–10 nm and 10–100 nm, respectively) reported for the Arctic atmosphere in May (Park et al., 2017) is a
prime example that biogenic DMS is a major contributor to NPF. A model study reported that DMS enhanced the mass of
sulfate particles in the size range 50–100 nm in regions north of 70° N (Ghahremaninezhad et al., 2019). During the bloom
and post-bloom periods a decline in anthropogenic sources and an increase in oceanic DMS source strength resulted in the
transition of major sulfate sources from Anth-$SO_4^{2-}$ to Bio-$SO_4^{2-}$, which highlights the increasing importance of biogenic
sulfur aerosols in the summer Arctic atmosphere. Biogenic organic aerosols in the high Arctic were reported to contribute
considerably to the concentrations of ultrafine and CCN particles from summer to early autumn when anthropogenic source
is lowest (Dall'Osto et al., 2017; Lange et al., 2019). Nonetheless, Anth-$SO_4^{2-}$ contributed considerably to the total $SO_4^{2-}$
budget during the post-bloom period, indicating that even in summer the Anth-$SO_4^{2-}$ transported from Europe or local
emissions can exert a significant influence on the sulfate budget in the Arctic atmosphere (Fig. S4) (Chen et al., 2016; Gogoi
et al., 2016; Dekhtyareva et al., 2018).

**4.2. Factors influencing the DMS oxidation pathways to either MSA or Bio-$SO_4^{2-}$ ($R_{Bio}$)**

**4.2.1. Seasonal variations in $R_{Bio}$**

    Our data spanning five years show two distinctive trends in $R_{Bio}$ among seasons or years. The first is that the values
of $R_{Bio}$ during the bloom and post-bloom periods (0.32 ± 0.15) were a factor of three higher than the pre-bloom values (0.09
± 0.07) (Fig. 5c). The large seasonal difference in $R_{Bio}$ could be explained by known factors including the concentration of
OH radicals (directly influenced by light intensity), air temperature (determining the oxidation pathway of DMS to either
MSA or Bio-$SO_4^{2-}$), and the chemical properties of existing particles (e.g., the black carbon concentration) (e.g., Saltzman et
al., 1986; Gondwe et al., 2004; Yan et al., 2020b). Among those, a major factor is the concentration of OH radicals. BrO
radicals also help facilitate the addition pathway in the oxidation of DMS, even at concentrations > 1 pptv level (von Glasow
and Crutzen, 2004). It has been hypothesized that the reactive bromines produced photochemically and heterogeneously at
sea ice and snowpack surfaces lead to the BrO enrichment over ice-covered regions (Abbatt et al., 2012; Fernandez et al.,
2019). Therefore, a high light intensity would favor the oxidation pathway of DMS to MSA, because this pathway is



effectively mediated by photochemically activated species including OH and BrO. The solar radiation ($51.3 \pm 36.1$ W m$^{-2}$) over the distant DMS source regions during the pre-bloom period was much lower than over the Greenland Sea and the Barents Sea during the bloom ($243.0 \pm 63.4$ W m$^{-2}$) and post-bloom ($222.5 \pm 70.5$ W m$^{-2}$) periods (Fig. 7). The low OH radical and reactive bromine concentrations during the pre-bloom period probably lowered the production of MSA from

DMS oxidation (i.e., weakening the addition pathway), and thereby resulted in the lower $R_{Bio}$ value ($0.09 \pm 0.07$) than was found during the bloom ($0.32 \pm 0.17$) and post-bloom ($0.32 \pm 0.13$) periods. Consequently, solar radiation was likely to be a major driver of the seasonal $R_{Bio}$ change in the Arctic atmosphere.

Another established factor that could affect the seasonal variations in $R_{Bio}$ is air temperature. At lower air temperatures DMS is oxidized more to MSA (leading to higher $R_{Bio}$ values), whereas at higher temperatures it is oxidized

more to Bio-SO$_4^{2-}$ (leading to lower $R_{Bio}$ values) (Hynes et al., 1986; Yin et al., 1990). For example, the values of $R_{Bio}$ measured near the equator (where the air temperature is high) are an order of magnitude lower than the values measured at high latitudes (where air temperature is low) (e.g., Bates et al., 1992b; Chen et al., 2012; Lin et al., 2012); these findings substantiate the linear dependence of the $R_{Bio}$ on the air temperature. Equally available is evidence against the temperature dependence of $R_{Bio}$ at given locations (Li et al., 1993; Legrand and Pasteur, 1998; Norman et al., 1999). In our study the air

temperature ($-1.8 \pm 2.2$ °C) in the distant DMS source regions (Region 2 in Fig. S10) during the pre-bloom period (March to April) was slightly lower than that ($-0.1 \pm 4.4$ °C) during the bloom and post-bloom periods (May to August) over the local Greenland and Barents seas (Region 1 in Fig. S10), where the higher $R_{Bio}$ values were more determined by the local DMS source. The empirical relationship between temperature and DMS oxidation products points to lower $R_{Bio}$ values at higher air temperatures (MSA/SO$_4^{2-}$ (%) = $-1.5 \times$ temperature (°C) + 42.2; Bates et al., 1992b). In contrast, our measurements pointed

to the opposite trend, and further indicates that factors other than air temperature might be more important in the partitioning of DMS into MSA or Bio-SO$_4^{2-}$. Similar to our results, in other high latitude studies light intensity has been reported to be more important than air temperature in determining the seasonal variations in $R_{Bio}$ (Gondwe et al., 2004).

The chemical properties of existing particles could influence the seasonal variations in $R_{Bio}$. In recent field studies the uptake of gaseous MSA onto particles was found to be sensitive to the chemical properties of those particles (Yan et al.,

2020b). In particular, hydrophobic and acidic particles in the atmosphere tended to hinder the adhesion of gaseous MSA to particles, while alkaline sea-salt particles tended to accelerate the adhesion process (Pszenny, 1992; Jefferson et al., 1998; Yan et al., 2020b). Elemental carbon particles emitted from fossil fuel combustion are highly hydrophobic, and sulfates in the aerosol particles are acidic. As a result, the air masses (rich in black carbon and sulfate) that originate from northern Europe would likely contain low MSA concentration in PM 2.5 particles, despite the fact that those air masses swept through the productive ocean areas during the pre-bloom period (Fig. 5a). In contrast, during the bloom period we found an elevation

of the MSA concentration, primarily as a result of two reinforcing processes: the greater DMS oxidation to MSA, and the enhanced condensation of gaseous MSA to the existing particles under less hydrophobic and acidic conditions. For each group of $R_{Bio}$ values, the lower concentrations of black carbon and sulfate resulted in the greater uptake of gaseous MSA,





and thereby resulted in higher $R_{Bio}$ values (Fig. 8). We also found significant inverse correlations between black carbon and
$R_{Bio}$ ($r$ = -0.79; Fig. 9a) and between total $SO_4^{2-}$ and $R_{Bio}$ ($r$ = -0.73; Fig. 9b); these tight correlations substantiate the
importance of the chemical properties of atmospheric particles in determining the rate of uptake of gaseous MSA by the
particles present in air. The number of samples measured during the bloom and post-bloom periods was higher in the groups
having large $R_{Bio}$ values (Fig. 8c). Therefore, the seasonal variations in $R_{Bio}$ measured at Ny-Ålesund were probably
controlled by the concentration of OH radicals (largely determined by light intensity) and the chemical properties of the
particles containing black carbon and sulfates.

The $R_{Bio}$ values in the present study, and those determined in other high latitude regions including Barrow in Alaska
(USA) and Neumayer station in the Antarctic coastal region, consistently pointed to the highest $R_{Bio}$ values occurring in
summer (Li et al., 1993; Legrand and Pasteur, 1998; Norman et al., 1999). We found that seasonal variability in $R_{Bio}$
measured in the Arctic region can be better explained by light conditions and the chemical properties of particles, rather than
by air temperature. Specifically, the $R_{Bio}$ values measured during the pre-bloom period poorly represent the oxidative
conditions of DMS in the Arctic atmosphere, because of the considerable intrusion of anthropogenic pollutants from the
distant northern Europe. Thus, the $R_{Bio}$ values measured during the bloom and post-bloom periods probably more accurately
represent the ratio of the oxidation products of DMS produced in the ocean regions surrounding Svalbard under the less
polluted conditions of the Arctic atmosphere.

**4.2.2 Interannual variations in $R_{Bio}$**

The second distinctive trend is the interannual difference in $R_{Bio}$. The $R_{Bio}$ values measured in 2017 were much
lower than the values in other years (2015, 2016, 2018, and 2019; Fig. 10). One explanation for large interannual variations
in $R_{Bio}$ is the difference in the condensation of gaseous MSA onto particles in the Arctic atmosphere. As noted above, the
chemical properties of particles largely determines the rate of MSA condensation onto them (Jefferson et al., 1998; Yan et
al., 2020b). During the pre-bloom and bloom periods in 2017, higher concentrations of black carbon and sulfate were found
relative to other years, and consequently, lower $R_{Bio}$ values found in 2017 (Fig. 10). However, we found no discernible
interannual difference in the concentrations of black carbon ($8.3 \pm 4.9$ ng m$^{-3}$ in 2017 and $9.5 \pm 6.1$ ng m$^{-3}$ in other years)
and total $SO_4^{2-}$ ($235 \pm 101$ ng m$^{-3}$ in 2017 and $232 \pm 134$ ng m$^{-3}$ in the other years) during the post-bloom period (Fig. 10).
To our surprise, during the post-bloom period the $R_{Bio}$ value ($0.22 \pm 0.07$) in 2017 was only half the rate measured in the
other years ($0.39 \pm 0.11$). The lack of association between black carbon and sulfate concentrations and the $R_{Bio}$ values
indicates that the chemical properties of particles may only partially impact the interannual variations in $R_{Bio}$ values
measured during the post-bloom period.

Air temperature difference may explain the interannual variations in $R_{Bio}$ during the post-bloom period (Fig. S11).
However, the lack of correlation (e.g. Bates et al., 1992b) we found between $R_{Bio}$ values and the mean temperatures of air
masses along the entire pathway to Svalbard (Fig. S11b) is consistent with the results of other studies (Savoie et al., 1992;


Legrand and Pasteur, 1998; Zhan et al., 2017), implying that variations in air temperature were not a driver of determining the DMS branching ratio. We also found no discernible differences in solar radiation and relative humidity between year of 2017 and other years, thus neither solar radiation nor relative humidity showed any association with $R_{Bio}$ (Fig. S11c–d). Thus, no meteorological factors adequately explain the interannual variations in $R_{Bio}$ during the post-bloom period. The

concurrent measurements of DMS and MSA during summer in the Southern Ocean reinforce our finding that temperature and relative humidity have negligible effects on the conversion of DMS to MSA (Yan et al., 2020a).

Our observation of lower $R_{Bio}$ values in 2017 than in other years may derive from a greater contribution of distant air masses (having low $R_{Bio}$ values) to the observation site. Air masses that have traveled greater distances are likely to have lower R values because MSA removal during the long-range transport exceeds that of Bio-$SO_4^{2-}$. However, we found no

discernable difference in the percentage fraction of distant air masses arriving at the site in 2017 versus in other years, arguing against this hypothesis. In a similar vein, differences in $R_{Bio}$ may also arise for local air masses within different airborne periods. This would be expected to apply to the local air masses over the Greenland and Barents Sea near the observation site. Because MSA (which contributes more to the growth of existing particles) and Bio-$SO_4^{2-}$ (which contributes more to new particle formation) have different fates in the atmosphere, local air masses with different airborne

periods would be expected to have different $R_{Bio}$ values. This is another possible cause of the observed interannual variations in R during the post-bloom period. The interannual variations in $R_{Bio}$ may also be associated with changes in the heterogeneous oxidation of MSA to sulfate by OH radicals on wet aerosol particles or cloud droplets (Hoffmann et al., 2016; Mungall et al., 2018), although assessment of this possibility was beyond the scope of the present study.

As sulfate and MSA particles have different roles in terms of particle formation and growth, the importance of $R_{Bio}$

is worth highlighting. Sulfate particles (including sulfuric acids) are known to produce 4–6 times more submicron sized particles than MSA, leading to a 10-fold stronger cooling effect via scattering of solar radiation (i.e., a direct effect), whereas the impacts of sulfate and MSA particles on cloud microphysics (i.e., an indirect effect) are comparable (Hodshire et al., 2019). Our results showing considerable seasonal or interannual variations in $R_{Bio}$ indicate that the use of a single ratio for determining the oxidation products of DMS and evaluating the contribution of biogenic sources to the total sulfur budget at

particular locations is problematic.

## 5. Conclusion and Implication

This study shows that in the Arctic atmosphere extensive production of the oxidation products of DMS (i.e., Bio-$SO_4^{2-}$ and MSA) occurred from the onset to the termination of phytoplankton blooms between 2015 and 2019. Anth-$SO_4^{2-}$

was found to be the largest contributor to total sulfate aerosols during the pre-bloom periods, as a result of the influence of Arctic haze. Its contribution was comparable to that of Bio-$SO_4^{2-}$ during the bloom and post-bloom periods. We also found large interannual variations in anthropogenic and biogenic sulfur aerosols. Moreover, the ratio of MSA to Bio-$SO_4^{2-}$ ($R_{Bio}$)





tended to be higher ($0.32 \pm 0.15$) in summer than in early spring ($0.09 \pm 0.07$), indicating that in summer only 30 % of the oceanic DMS was oxidized to MSA, and the remainder was oxidized to Bio-$SO_4^{2-}$ aerosols. Our results imply that NPF, and

subsequent growth of those particles to form CCN, are governed by both Bio-$SO_4^{2-}$ and MSA when $R_{Bio}$ is high (bloom and post-bloom periods), but that when $R_{Bio}$ is low (pre-bloom period) MSA makes only a small contribution to particle growth and other molecules with low-volatility vapors (e.g., highly oxygenated organic molecules) are more involved in particle growth near Svalbard. The large interannual variability of $R_{Bio}$ further indicates that condensational growth following NPF can be affected by MSA or other molecules with low-volatility vapors, depending on the branching ratio of DMS oxidation.

415       In modelling studies (Vallina et al., 2006, 2007) the annual contribution of biogenically induced CCN to total global CCN has been estimated to be > 30 %, and up to 80 % in the austral summer in the Southern Ocean. This is similar to findings for the Northern hemisphere, where Bio-$SO_4^{2-}$ particles accounted for > 60 % of CCN in late spring (May and June) in the North Atlantic (Sanchez et al., 2018). An acceleration of sea ice retreat and an increase in melt ponds in the Arctic Ocean will increase biogenic DMS production, resulting in a greater contribution of biogenic S aerosols to atmospheric

aerosol formation and climate regulation (Arrigo et al., 2008; Gourdal et al., 2018; Park et al., 2019). Our measurements primarily focused on the particle phase of sulfur species (particles < 2.5 μm), but did not cover the initial phase of DMS oxidation and particle growth (i.e. nano size scales), including the gas-phase composition of sulfur species. Therefore, the integrated study of both the gas and particle phases of sulfur compounds (including gaseous MSA, $SO_4^{2-}$, and hydroperoxymethyl thioformate), ocean colors, and sea ice properties will help define the climate-relevant impacts of

oxidation products of biogenic DMS in the Arctic environment.





**Data availability**

All data needed to draw the conclusions in the present study are presented in this report and/or the Supplementary Materials. For additional data related to this study, please contact the corresponding author (Kitack Lee; ktl@postech.ac.kr).

**Supplement**

The supplement related to this article is available on line at https://

**Author contributions**

S.J., K.P., Y.Y., and K.L. designed the data analysis and wrote the manuscript. S.J. and K.P. performed the data evaluation and analyses. K.K. and H.C. performed the ion chromatograph measurements. K.E. provided the black carbon data. R.T. and
S.B. were involved in aerosol sample collection. B.L., R.K., and O.H. contributed to the interpretation of the results.

**Competing interests**

The authors declare that they have no conflict of interest.

**Acknowledgements**

This research was supported by National Research Foundation of Korea funded by the Ministry of Science and ICT (Basic
Research Program; NRF-2020R1A4A1018818) and KOPRI-PN20081 (CAPEC project; NRF-2016M1A5A1901769).



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

Figure 1: (a) Location of the aerosol sampling site (black pentagram; Gruvebadet observatory; 78.9° N, 11.9° E) and the ocean domains (70° N–80° N, 25° W–50° E for Region 1; 50° N–70° N, 25° W–50° E for Region 2) defined for this study. Mean Chl-*a* concentration for (b) March and April, (c) May and June, and (d) July and August over the period of 2015–2019, overlaid with air mass trajectory clusters that represent the dominant pathways of air masses reaching to the observation site.






Figure 2: (a) Atmospheric DMS mixing ratios measured at Zeppelin station, Svalbard, in 2015, 2016, 2017, 2018, and 2019. (b) Stable isotope composition of sulfate aerosols. Three end-member values; $\delta^{34}S_{SS} = 21 \pm 0.1$ ‰ for sea-salt sulfates; $\delta^{34}S_{Anth} = 5 \pm 1$ ‰ for anthropogenic sulfate; and $\delta^{34}S_{Bio} = 18 \pm 2$ ‰ for biogenic sulfates.




Figure 3: Monthly data during the measurement years (2015-2019) for: (a) black carbon (BC), (b) atmospheric DMS mixing ratio, (c) sulfur isotope measurements ($\delta^{34}S$), (d) NSS-SO$_4^{2-}$, (e) Anth-SO$_4^{2-}$, (f) Bio-SO$_4^{2-}$, (g) MSA, (h) Bio-aerosol, and (i) MSA to Bio-aerosol ratio (R$_{Bio}$) during the measurement years (2015–2019). Solid lines and red crosses represent the median and mean values of the data, respectively.



Figure 4: Aerosol concentrations for: (a) NSS-SO$_4^{2-}$ (total SO$_4^{2-}$ minus ss-SO$_4^{2-}$); (b) Anth-SO$_4^{2-}$, and (c) Bio-SO$_4^{2-}$. The colored solid lines indicate 15-day moving average values.


Figure 5: Aerosol concentrations of (a) MSA and (b) Bio-aerosol (MSA + Bio-SO$_4^{2-}$). (c) Variations in the ratio of MSA to Bio-aerosol (R$_{Bio}$). The colored solid lines indicate 15-day moving mean values.

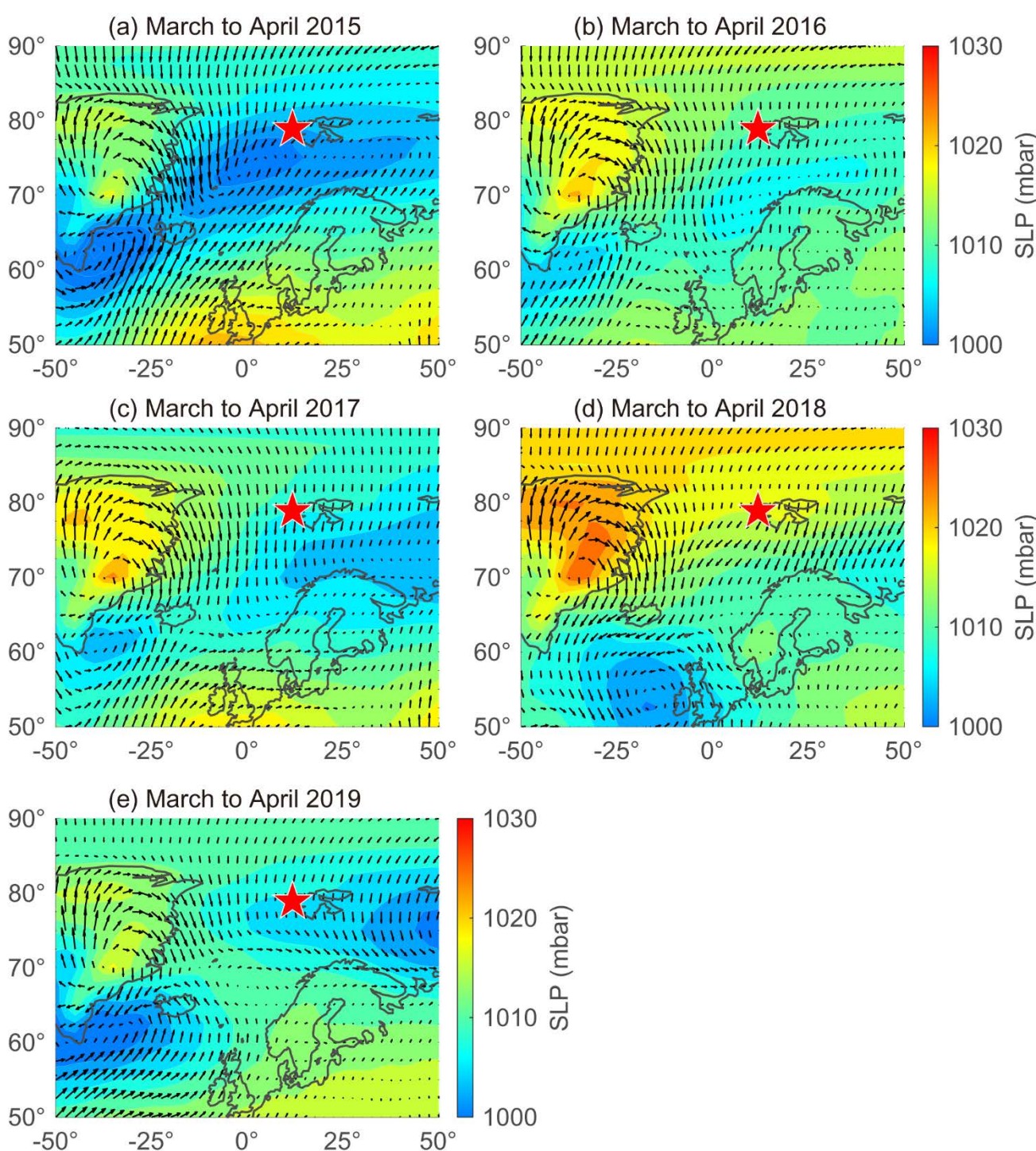

Figure 6: Sea level pressure (SLP) overlaid with wind vectors during March to April in (a) 2015, (b) 2016, (c) 2017, (d) 2018, and (b) 2019. Red stars indicate the location of the sampling site (Gruvebadet observatory; 78.9° N, 11.9° E).





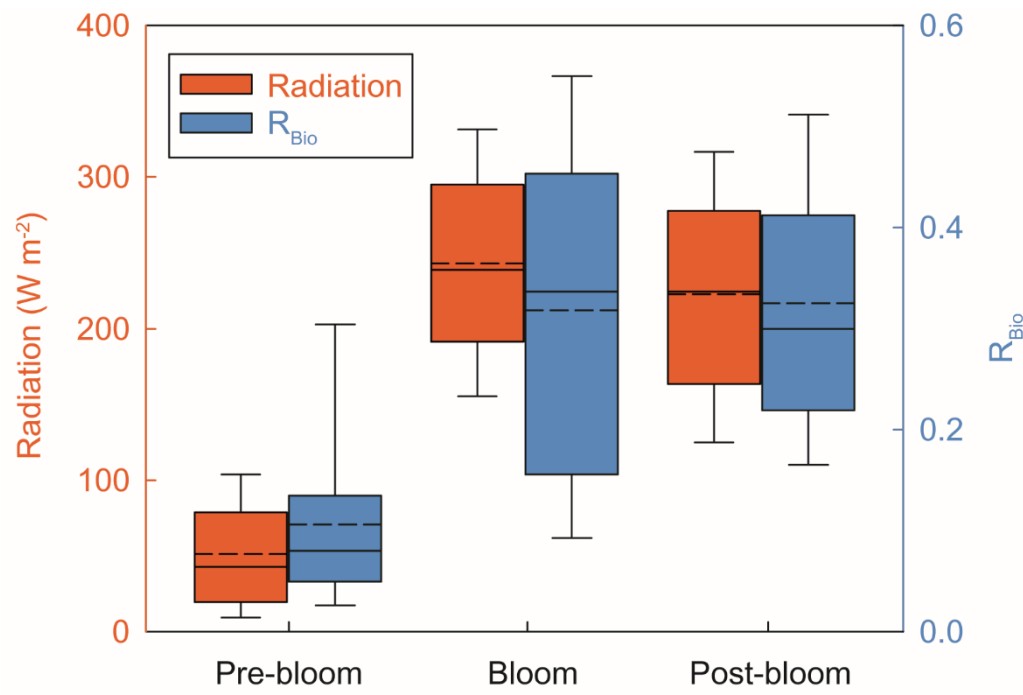

Figure 7: Five-year (2015, 2016, 2017, 2018 and 2019) mean radiation (red) and $R_{Bio}$ (blue) during the pre-bloom, bloom, and post-bloom periods. Solid line and dotted line represent median and mean value of each data in box plot, respectively.



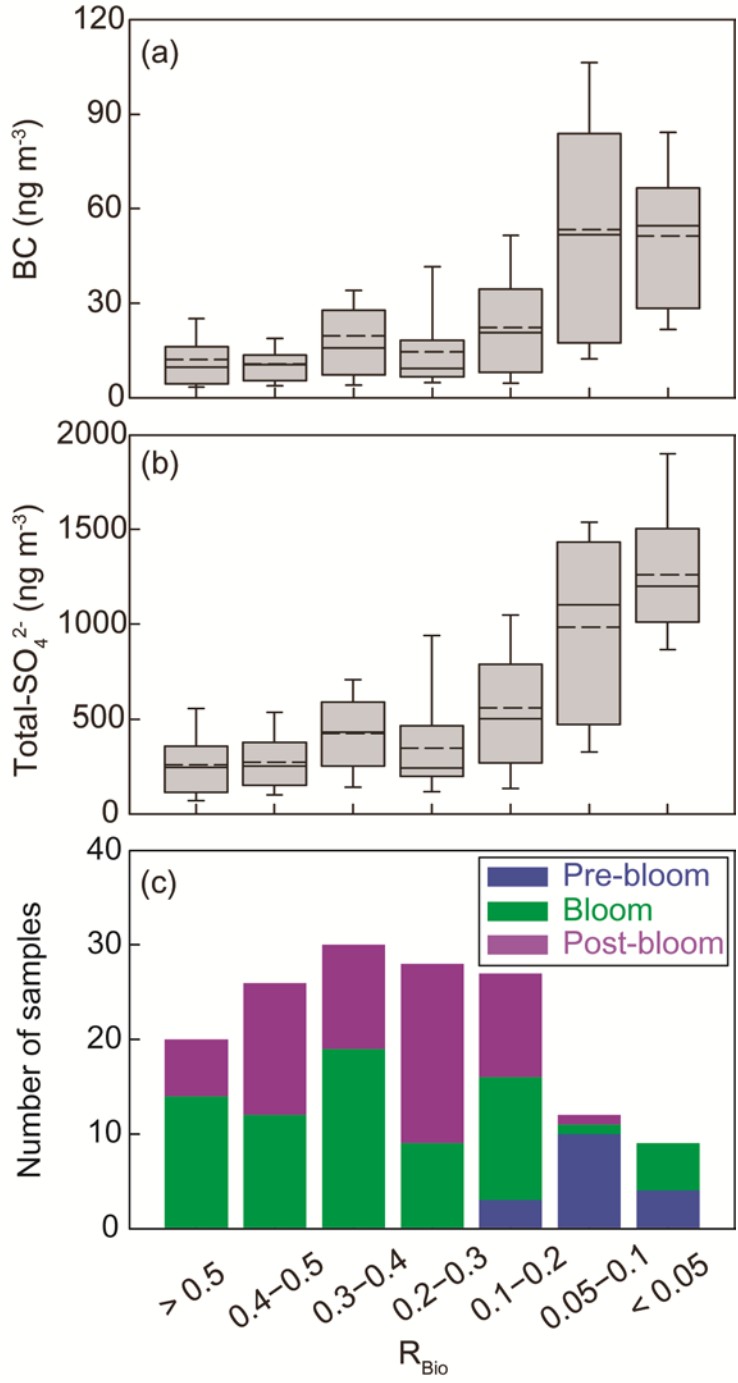

Figure 8: Plots of the seasonal (a) black carbon (BC) concentration versus $R_{Bio}$, (b) the total $SO_4^{2-}$ concentration versus $R_{Bio}$, and (c) the number of samples included in each $R_{Bio}$ group. The solid and dotted lines represent the median and mean values of the data in the box plots, respectively.




Figure 9: Scatter plot of (a) monthly mean black carbon (BC) concentration versus monthly mean $R_{Bio}$ and (b) monthly mean Total $SO_4^{2-}$ concentration versus monthly mean $R_{Bio}$. Error bars and black solid line represent $1\sigma$ and the best fit, respectively.



Figure 10: Concentration of (a–c) black carbon (BC), (d–f) total $SO_4^{2-}$, and (g–i) $R_{Bio}$ during pre-bloom, bloom, and post-bloom periods. The solid and dotted lines represent the median and mean values of the data in box plots, respectively.