# Peer review of "Large seasonal and interannual variations of biogenic sulfur compounds in the Arctic atmosphere (Svalbard; 78.9° N, 11.9° E)"

_Atmospheric Chemistry and Physics, 2020_

## Author Comment (AC1)

**Response to Referee 1**

We thank Referee 1 for providing insightful suggestions, which have considerably improved the readability of the revised manuscript. Our responses to general and specific comments (raised by Referee 1) are stated below.

**General comments**

**[Comment 1] This manuscript presents a 5-year time series of aerosol properties in spring and summer in the Atlantic Arctic (Gruvebadet station), focusing on the relative importance of natural and anthropogenic sources of sulfate and MSA in aerosols. The study presents an interesting dataset for the understanding of natural aerosol sources during the Arctic summer, which play an important climatic role. The article is generally well written and concise and the measurements reported look technically sound. However the study does not present, to my understanding, relevant conceptual or methodological innovations. In particular, the factors that drive the transition between the Arctic haze season, when anthropogenic pollution dominates sulfate aerosols, and the "clean" summer season, when local biological sources play a more important role, have been known for a long time (as demonstrated by many of the references cited by the authors). This pattern results from the interplay between the seasonal changes in atmospheric circulation, ocean activity and atmospheric photochemistry and condensation sink. To make the article less incremental and more interesting, in my view the authors should place more emphasis on the intriguing interannual variations, and less on the well-known seasonal shift from anthropogenic to natural aerosols. Interannual changes are, in my view, the most interesting aspect of the dataset, but the authors fail at explaining their causes. A more complete analysis of satellite data, including the use of recently developed satellite algorithms for marine sulfur compounds, combined with air mass back trajectories, could help explain interannual differences. Satellite and air-mass datasets are presented but not fully exploited:**

[Response 1] To thoroughly explore possible cause(s) for interannual variations in the concentrations of MSA and Bio-SO$_4^{2-}$ and the MSA to Bio-S-aerosol ratio (which were not explored in the original manuscript), we carried out additional analysis of ocean chlorophyll and air mass back trajectory data. In particular, results from this analysis indicate that interannual (as well as seasonal) variations in R$_{Bio}$ values were largely explained by variations in the air mass retention time over the DMS source regions. To support this new finding we have added to the revised manuscript three new figures (Figures 9, 10 and 11) along with relevant explanations (lines 175–183, 371–382, and 414–419).

All relevant modifications (three figures and three paragraphs) made were directly copied from the revised manuscript:

Lines 175–183: "*The retention time for air masses in each domain type (including the ocean, marginal ice zone, multi-year ice, and land) was calculated based on the sea ice index at 25-km resolution provided by the National Snow and Ice Data Center (Choi et al., 2019). Note that the marginal ice zone and multi-year ice represent the areas in which the sea ice cover is 15–80% and > 80%, respectively (Stroeve et al., 2016). The air mass exposure to chlorophyll (E$_{Chl}$) was calculated to estimate the biological exposure history of air masses arriving at the observation site (Arnold et al., 2010; Park et al., 2018), according to Equation 5:*

$$E_{Chl} = \frac{\sum_{t=1}^{120} Chl}{n} \quad (5)$$

*where Chl is the 8-day mean Chl-a concentration within a radius of 25 km at a given time point (t) along the 5-day air mass back trajectory, and n is the total number of time points for which valid Chl-a values were available.*"

Lines 371–382: *"A strong positive correlation between monthly mean $R_{Bio}$ and the air mass exposure to chlorophyll ($E_{Chl}$) was observed during the study period ($r = 0.82$). The retention time of air masses over the ocean and marginal ice zone (i.e., DMS source regions) was also positively correlated with $R_{Bio}$ values ($r = 0.54$). The $R_{Bio}$ values decreased with decreasing air mass retention time over the land and multi-year ice regions (i.e., the non-DMS-source regions). The concentration of MSA was positively correlated with the mean Chl-a concentration in areas surrounding the observation site, but no similarly clear correlation was found between Bio-$SO_4^{2-}$ and Chl-a (Fig. S9). The absence of a correlation between Bio-$SO_4^{2-}$ and Chl-a indicates that the concentration of Bio-$SO_4^{2-}$ measured at the observation site included sulfur compounds produced locally and in distant regions, because the greater atmospheric residence time of Bio-$SO_4^{2-}$ (relative to MSA) indicates greater intrusion of Bio-$SO_4^{2-}$ into the observation site. Hence, air masses that have been extensively exposed to local biological activities are likely to have higher $R_{Bio}$ values. Therefore, the seasonal variations in $R_{Bio}$ measured at Ny-Ålesund were probably controlled by the concentration of OH radicals (largely determined by light intensity), the chemical properties of the particles containing black carbon and sulfates, and biological activities surrounding the observation site."*

Lines 414–419: *"Analysis of air mass back-trajectory data indicated that the air mass exposure to chlorophyll ($E_{Chl}$) in 2017 ($0.44 \pm 0.21$) was 30% lower than in other years ($0.63 \pm 0.35$). The mean retention time of air masses over the sea ice and land areas (i.e., non-DMS source regions) in 2017 ($40.9 \pm 27.9$ h) was 25% longer than that estimated for other years ($32.3 \pm 19.2$ h), whereas the mean retention time of air masses over the ocean and marginal ice regions (i.e., the DMS source regions) was lower in 2017 ($79.1 \pm 27.9$ h) than in other years ($87.7 \pm 19.2$ hours) (Fig. 11). Hence, the 2017 $R_{Bio}$ values were 40% lower than those in 2018 and 2019, probably because more air masses swept over non-DMS source regions."*

[Figure]

Figure 9: Scatter plots of monthly mean $R_{Bio}$ values as a function of: (a) the monthly mean black carbon (BC) concentration; (b) the monthly mean total $SO_4^{2-}$ concentration; (c) the air mass retention time over the ocean and the marginal ice zone (MIZ); (d) the air mass retention time over multi-year ice and land areas; and (e) the monthly mean air mass exposure to chlorophyll ($E_{Chl}$). Error bars and the black solid line represent 1σ and the best fit, respectively.

[Figure]

Figure 10: $R_{Bio}$ (a–c) and the black carbon (BC) and total $SO_4^{2-}$ concentrations (d–f) during pre-bloom, bloom, and post-bloom periods. Error bars represent $1\sigma$.

[Figure]

Figure 11: Air mass exposure to chlorophyll ($E_{Chl}$) (a–c) and the air mass residence times over the ocean and marginal ice zone (MIZ) and the multi-year ice and land areas (d–f) during pre-bloom, bloom, post-bloom periods. Error bars represent $1\sigma$.

**[Comment 2] In addition, I suggest a more careful consideration of differential sources and sinks of MSA and SO$_4$ in the Discussion, because both sources and sinks modulate the MSA/Bio-aerosol ratio:**

[Response 2] In the revised Discussion (see lines 371–382, and 414–419), we have carefully evaluated all possible sources (or factors) for causing variations of the MSA and SO$_4$ concentrations in order to identify key factor(s) that dominantly influence the interannual variations in the MSA/Bio-S-aerosol ratio. Considering all possibilities, we found that the magnitude of MSA uptake by black carbon and total $SO_4^{2-}$ explained the difference in the MSA/Bio-S-aerosol ratio between the prebloom versus bloom and post-bloom periods (low values in the pre-bloom versus high values in the bloom and post-bloom period), whereas the magnitude of air mass exposure to the ocean DMS source determined the interannual variations in the MSA/Bio-S-aerosol ratio during the post-bloom periods ($R_{Bio}$ = 0.22 in 2017 versus 0.39 in other years).

**[Comment 3] Less importantly, I prompt the authors to word more carefully some sentences on marine and sea-ice biological activity:**
[Response 3] Because our measurements presented here cannot adequately prove or disprove that sea-ice algae is a large source of DMS, we did not rule out the possibility that the sea-ice algae is an important DMS source. The revised sentence reads *"DMS is only produced in the upper ocean by means of multiple biological processes"* to *"DMS is produced through multiple biological processes occurring in pelagic and sympagic ecosystem"* (lines 70–71). We have also changed *"despite the absence of biological activity in the sea ice-covered oceans surrounding Svalbard"* to *"despite low biological activity (as indicated by DMS mixing ratios of < 10 pptv)"* (lines 272–273).

   **Specific comments**

**[Comment 4] Line 26. Please tone down. Replace "obviously" by something more neutral. Can sea-ice DMS sources by completely ruled out?**
[Response 4] We agree with this referee that sea ice algae could act as a significant source of atmospheric DMS in the Arctic atmosphere. Therefore, we have toned down the sentence by deleting *"obviously"* and modifying the sentence as follows (lines 26–28): *"These probably originated in regions to the south (the North Atlantic Ocean and the Norwegian Sea), rather than in ocean areas in the proximity of Ny-Ålesund."*

**[Comment 5] Line 42. Acidification appears out of the blue here and breaks the flow. If the authors want to elaborate on the potential impact of acidification on marine DMS emission (for which there is inconclusive evidence), as I suspect, this needs to be better introduced:**
[Response 5] We described the potential impact of ocean acidification on marine DMS emission in future environment (lines 46−49) as follows: *"Moreover, acidification of the Arctic Ocean has been enhanced because of the increasing addition of anthropogenic $CO_2$, facilitated by ocean freshening and greater air-sea $CO_2$ exchange (Lee et al., 2011); and ocean acidification potentially impacts on the net production and fluxes of marine trace gases, and so affects climate (Hoppkins et al., 2020)."*

**[Comment 6] Line 50. Does this conform with more up-to-date references on MSA? Veres et al. 2020; Hoffmann et al. 2016; Dawson et al. 2012 (the three in PNAS). Please review references on MSA chemistry in the rest of the Introduction:**
[Response 6] We have revised the roles of DMS-derived particles in the formation and growth of aerosol particles in lines 52−59. We have also added a short paragraph (lines 59−63) that explains the recent findings associated with the formation of MSA and its reactions in the atmosphere. Note that we have added to the revised manuscript (lines 59−63) relevant references for the formation of MSA and its reactions (Veres et al., 2020; Chen et al., 2015; 2016; Dawson et al., 2012; Bork et al., 2014; Hoffmann et al., 2016; Yan et al., 2020).

All relevant modifications (two paragraphs) made were directly copied from the revised manuscript:

Lines 52−59: *"Sulfurous compounds including $SO_2$, methanesulfonic acid, and hydroperoxymethyl thioformate in the atmosphere are the oxidation products of dimethyl sulfide (DMS). These effectively form new particles through homogeneous nucleation and clustering reactions that are closely linked to water vapor and ammonia (negative ion-induced ternary nucleation), and contribute to particle growth (Kulmala, 2003; Kulmala et al., 2004; Veres et al., 2020). Sulfuric acid is widely recognized as a driver of new particle formation (NPF) (Kulmala, 2003), whereas methanesulfonic acid (MSA) particles tend to condense onto particles that are already present (existing particles), and so contribute to particle growth (Wyslouzil, et al., 1991; Leaitch et al., 2013; Hayashida et al., 2017)."*

Lines 59−63: *"However, recent studies have provided evidence for MSA involvement in new particle formation; for example, the reaction of MSA with amines or ammonia in the presence of water results in particle formation and growth (Dawson et al., 2012; Chen et al., 2015; 2016a). MSA also indirectly contributes to NPF by enhancing the formation of $H_2SO_4$-amines clusters (Bork et al., 2014). Some studies have reported that MSA only increased the mass of particles and not their number (Hoffmann et al., 2016; Yan et al., 2020), suggesting a minor role for MSA in NPF."*

**[Comment 7] Line 61. Please mention sea ice, where DMS production and emission does also occur. Levasseur et al. 2013 (NatGeo); Park et al. 2019 (ESPI); etc.:**
[Response 7] We have already addressed in our Responses 3 and 4. We have explicitly stated in lines 70–71 that the sea-ice algae can be a significant source for atmospheric DMS. Relevant references (Levasseur, 2013; Park et al. 2019) were also added to line 72.

**[Comment 8] Line 83. Is "Bio-aerosol" an appropriate expression? It seems to disregard non-sulfur biogenic aerosol sources, like VOCs other than DMS and primary organic aerosol. I suggest using a more precise expression:**
[Response 8] We have changed "*biogenic aerosols*" to "*biogenic S aerosols*" and "*Bio-aerosol*" to "*Bio-S-aerosol*".

**[Comment 9] Line 168. "a" or "the"?**
[Response 9] We have replaced "*a*" to "*the*" (line 186).

**[Comment 10] Line 171. For clarity, please add "during previous studies" before "Svalbard":**
[Response 10] We have added "*As previously confirmed in other studies (e.g. Arnold et al., 2010; Park et al., 2013; Mungall et al., 2016)*" to lines 187−188.

**[Comment 11] Line 190. This is not that surprising. Norman et al. (1999, JGR) already showed a dominant contribution of Anth-SO4 all year round at Alert (high Canadian Arctic), with the lowest monthly Anth contribution in August with about 55% (roughly corresponding to 45% DMS contribution). The authors may also want to check Mahmood et al. 2019 (ACP), which compared that dataset to model outputs which showed agreement:**
[Response 11] Since the large anthropogenic contribution in summer was found in earlier studies, we have deleted "*It is surprising that*". The revised sentence reads "……… *which was consistent with the previous findings*" (lines 208−210). We have added to line 210 relevant references that report the contribution of anthropogenic sulfate to total sulfate burden in the Arctic atmosphere (Li and Barrie, 1993; Norman et al., 1999; Udisti et al., 2016).

**[Comment 12] Line 201. Please replace "nearly absent" by something more objective, like a Chl a concentration range:**
[Response 12] We have now added an exact Chl-*a* concentration to line 221. The revised sentence reads "………*the chlorophyll-a concentration remained lower than 0.5 mg m$^{-3}$*".

**[Comment 13] Line 252. "absence of biological activity": Even if back-trajectories do not support a sea-ice source, this information needs to be corrected because sea ice can host extremely active microbes (Leu et al. 2015, PiO) which can produce DMS in significant amounts, e.g. Levasseur et al. 2013 (NatGeo); Hayashida et al. 2020 (GBC):**
[Response 13] We agree with this referee that sympagic ecosystem can be a hotspot for DMS emission. Thus, we have changed "*despite the absence of biological activity in the sea ice-covered oceans surrounding Svalbard*" to "*despite low biological activity (as indicated by DMS mixing ratios of < 10 pptv)*" (lines 272−273).

**[Comment 14] Line 280. The conclusions of the study of Park et al. 2018, quoted here, relied on a satellite proxy for DMSP-producing phytoplankton. Given that the satellite algorithm was**

consistent with atmospheric measurements, why not using it again, in combination with air mass back-trajectories, to understand DMS source regions in the current study?

[Response 14] In our response to the general Comment 1, we have added sentences (lines 175–183, 371–382, and 414–419) and figures (Figs. 9, 10 and 11) to describe and show factors that are responsible for seasonal and interannual variations in $R_{Bio}$.

**[Comment 15] Line 340. Moffett et al. 2020 (JGR-A) suggested MSA condensation on anthropogenic (fossil fuel combustion) particles. Please revise if needed:**

[Response 15] We have added the statement to lines 358–360 indicating this condensation mechanism of MSA onto anthropogenic particles. The added sentence reads "*However, only a small proportion of the anthropogenic particles formed in the polluted coastal and urban sites was found to be associated with MSA (Gaston et al., 2010; Yan et al., 2020) formed from the oxidation of aqueous DMS catalyzed by iron and vanadium (Gaston et al., 2010; Moffett et al., 2020)*".

**[Comment 16] Line 370. Doesn't this conclusion contradict previous paragraphs? (eg L340):**

[Response 16] We found two distinct factors: one influencing *the interannual variations in the $R_{Bio}$ during the pre-bloom and bloom periods*; and the other influencing *$R_{Bio}$ values measured during the post-bloom period.* In the former (pre-bloom and bloom periods), chemical properties of the existing particles were responsible for the interannual variations in the $R_{Bio}$, whereas in the latter (post-bloom periods) another factor (i.e., air masses retention time over the DMS source regions) played more important role in determining $R_{Bio}$ variations. The revised conclusion (lines 403−404) now read "*factors other than chemical properties of existing particles affected the interannual variation in $R_{Bio}$ values measured during the post-bloom period.*".

**[Comment 17] Line 375. Please check Moffett et al. 2020 (JGR-A) for time series of MSA and nss SO4 in the Pacific sector of the Arctic (Utqiaġvik, station formerly known as Barrow):**

[Response 17] Moffett et al. (2020) was added to line 408.

**[Comment 18] Line 398. Is the use of a "single ratio" common practice in atmospheric chemistry modelling studies? Please provide references:**

[Response 18] We have changed "*the use of single ratio*" to "*the conventional approach of using asymptotic values*" in lines 424−425 and added to line 426 relevant references (Udisti et al., 2012 and 2016; Norman et al., 1999).

**[Comment 19] Line 408. Can we really assume that the MSA/Bio-aerosol ratio is equal to the branching ratio, without knowing the differences in the sinks? Concentrations in aerosols results from both sources and sinks, which are very likely different for each compound. Please revise:**

[Response 19] We have deleted "*indicating that in summer only 30% of the oceanic DMS was oxidized to MSA, and the remainder was oxidized to Bio-$SO_4^{2-}$ aerosols*" because the measured $R_{Bio}$ does not directly represent the branching ratio of DMS oxidation without knowing the differences in the sinks.

**[Comment 20] Line 420. Please check Gali et al. 2019 (PNAS), which seems a relevant reference to support this point:**

[Response 20] The work of Galí et al (2019) was added to line 445−446.

**[Comment 21] Line 266. "snow", not plural:**

[Response 21] We have changed "*snows*" to "*snow*" line 286.

**References cited in our response to Referee #1's comments (most references listed below were cited in our revised manuscript)**

[revised manuscript text omitted]

---

## Author Comment (AC2)

**Response to Referee 2**

We thank Referee 2 for providing valuable suggestions that have improved the readability of our revised manuscript. Our responses to this Referee's comments are provided below.

**Comments on abstract**

**[Comment 1] The abstract summary seems to concentrate on the MSA findings/impacts e.g. growth of new particles to CCN yet it is stated later that a more significant impact could be on the formation of Bio-$SO_4^{2-}$ which potentially impacts more on new particle formation and therefore has a direct 10-fold impact on cooling. Is this less significant in this study because the absolute amounts of Bio-$SO_4^{2-}$ aerosol are small compared to the MSA aerosol? It is unclear from the $R_{Bio}$ ratio how significant this relative contribution is and whether it should be stated a bit more in the abstract?**

[Response 1] The concentration of Bio-$SO_4^{2-}$ was 3 to 10 times higher than that of MSA during the study period. The concentration of MSA is typically used as an indicator of DMS-derived particles. This is because MSA is exclusively formed by the oxidation of DMS, while sulfate is of multiple origins (including DMS, sea-salt, and anthropogenic emission) which are not possible to distinguish without knowing S-isotope information and ion concentration data. Thus, literature ratios of MSA to Bio-$SO_4^{2-}$ are typically used to calculate the total amounts of DMS-derived aerosols (Udisti et al., 2012 and 2016; Norman et al., 1999).

We have added a short paragraph (lines 28–29) indicating the importance of this ratio in the abstract: The added statements read *"Another oxidation product of DMS is MSA, and the ratio of MSA to Bio-$SO_4^{2-}$ is extensively used to estimate the total amount of DMS-derived aerosol particles in remote marine environments."*. We also added *"MSA is not a conservative tracer for DMS-derived particles"* and deleted *"to a size at which they could act as condensation nuclei"* (lines 40–41).

**Specific comments**

**[Comment 2] Line 23. How can 50% of the NSS-$SO_4^{2-}$ be Anth-$SO_4^{2-}$? Do you mean it was produced from it? Rephrase:**

[Response 2] NSS-$SO_4^{2-}$ is a sum of Anth-$SO_4^{2-}$ and Bio-$SO_4^{2-}$. This sentence appeared to be misleading. So we have changed *"NSS-$SO_4^{2-}$"* to *"NSS-$SO_4^{2-}$ (sum of Anth-$SO_4^{2-}$ and Bio-$SO_4^{2-}$)"* for clarity (line 23).

**[Comment 3] Line 151. Reference needed for sources of black carbon from fossil fuel, burning?:**

[Response 3] Relevant literatures have been cited (Chen et al., 2016; Massling et al., 2015) (line 161).

**[Comment 4] Section 3.4. Could a table be included to summarise the $R_{Bio}$ values in different conditions/air masses, maybe include the temperature/light intensity if dependence is interesting (with reference to lines 320–333)?**

[Response 4] As this referee suggested, we have added a new table (Table S2) that summarizes multiple environmental variables that could affect $R_{Bio}$.

**Table S2.** Summarized seasonal temperature, solar radiation, and $R_{Bio}$

|  | Pre-bloom | Bloom | Post-bloom |
|---|---|---|---|
| Temperature (°C) | -1.8 $\pm$ 2.2 | -3.4 $\pm$ 3.4 | 3.3 $\pm$ 1.5 |
| Radiation (W m$^{-2}$) | 51.3 $\pm$ 36.1 | 243.0 $\pm$ 63.4 | 222.5 $\pm$ 70.5 |
| $R_{Bio}$ | 0.09 $\pm$ 0.07 | 0.32 $\pm$ 0.17 | 0.32 $\pm$ 0.13 |

**[Comment 5] Line 288. Missing 'hyphens' in brackets:**
[Response 5] We have hyphens in all relevant places in the brackets (line 308).

**[Comment 6] Figure 5b. Set max y-axis scale to 1500?:**
[Response 6] We have modified the y-axis scale of Figure 5b.

[Figure]

Revised Figure 5b: Aerosol concentration of Bio-S-aerosol (MSA + Bio-SO$_4^{2-}$). The colored solid lines indicate 15-day moving mean values.

**[Comment 7] Line 380. R$_{Bio}$, rather than just R:**
[Response 7] We have changed "$R$" to "$R_{Bio}$".

**Comments on conclusions**

**[Comment 8] Since the concentration of OH plays such a big role in the DMS oxidation, is there anything that can be said about the climatic potential of the effect of increasing or decreasing OH concentrations over time on these findings (e.g. increasing global methane could lead to decreasing OH). Is there any potential trend over the years or just interannual variability?**
[Response 8] We believe that interannual variation of OH radical may not seriously change the DMS oxidation pathway at this high Arctic site during our study period. Nevertheless, future studies on the climatic function of reactive oxidants are necessary to understand chemical oxidation process of biogenic DMS. Thus, we have added a short paragraph that emphasizes the potential impact of key oxidants including OH on DMS oxidation process as follows (lines 446–448): *"Another important factor that may be involved in the formation of biogenic CCN is changes in the atmospheric concentrations of OH, NO$_x$ and BrO; these are likely to be affected by future climate change and increasing anthropogenic perturbations (e.g., sea ice decline, increasing reduced carbon emissions) (Alexander and Mickley, 2015)."* and we have replaced *"including the gas-phase composition of sulfur species"* to *"including the concentration of key oxidant and gas-phase composition of sulfur species"* (lines 450–451).

**[Comment 9] I agree that more work should include integration with DMS, is this really beyond the scope of this paper?**
[Response 9] Our measurements primarily focused on the seasonal and interannual variations in DMS-derived particles. Future studies are required to define the climatic roles of DMS-derived particles with comprehensive and simultaneous physiochemical properties of aerosol particles, and its precursor compounds.

**References cited in our response to Referee #2's comments (most references listed below were cited in our revised manuscript)**
Alexander, B., and Mickley, L. J.: Paleo-perspectives on potential future changes in the oxidative capacity of the atmosphere due to climate change and anthropogenic emissions, Current Pollution Reports, 1(2), 57–69, https://doi.org/10.1007/s40726-015-0006-0, 2015.

Chen, L., Li, W., Zhan, J., Wang, J., Zhang, Y., and Yang, X.: Increase in Aerosol Black Carbon in the 2000s over Ny-Ålesund in the Summer, J. Atmos. Sci., 73(1), 251–262, https://doi.org/10.1175/JAS-D-15-0009.1, 2016.

Massling, A., Nielsen, I. E., Kristensen, D., Christensen, J. H., Sørensen, L. L., Jensen, B., Nguyen, Q. T., Nøjgaard, J. K., Glasius, M. and Skov, H.: Atmospheric black carbon and sulfate concentrations in Northeast Greenland, Atmos. Chem. Phys., 15(16), 9681–9692, https://doi.org/10.5194/acp-15-9681-2015, 2015.

Norman, A. L., Barrie, L. A., Toom-Sauntry, D., Sirois, A., Krouse, H. R., Li, S. M., and Sharma, S.: Sources of aerosol sulphate at Alert: Apportionment using stable isotopes, J. Geophys. Res.-Atmos., 104, 11619–11631, https://doi.org/10.1029/1999JD900078, 1999.

Udisti, R., Dayan, U., Becagli, S., Busetto, M., Frosini, D., Legrand, M., Lucarelli, F., Preunkert, S., Severi, M., Traversi, R., and Vitale, V.: Sea spray aerosol in central Antarctica. Present atmospheric behaviour and implications for paleoclimatic reconstructions, Atmos. Environ., 52, 109–120, 2012.

Udisti, R., Bazzano, A., Becagli, S., Bolzacchini, E., Caiazzo, L., Cappelletti, D., Ferrero, L., Frosini, D., Giardi, F., Grotti, M., Lupi, A., Malandrino, M., Mazzola, M., Moroni, B., Severi, M., Traversi, R., Viola, A., and Vitale, V.: Sulfate source apportionment in the Ny Ålesund (Svalbard Islands) Arctic aerosol, Rend. Lincei, 27, S85–S94, https://doi.org/10.1007/s12210-016-0517-7, 2016.